# Anti-Survival Effect of SI306 and Its Derivatives on Human Glioblastoma Cells

**DOI:** 10.3390/pharmaceutics14071399

**Published:** 2022-07-01

**Authors:** Lorenzo Monteleone, Barbara Marengo, Francesca Musumeci, Giancarlo Grossi, Anna Carbone, Giulia E. Valenti, Cinzia Domenicotti, Silvia Schenone

**Affiliations:** 1Department of Experimental Medicine (DIMES), General Pathology Section, University of Genoa, 16132 Genoa, Italy; lorenzomonteleone94@gmail.com (L.M.); barbara.marengo@unige.it (B.M.); giuliaelda.valenti@edu.unige.it (G.E.V.); 2Inter-University Center for the Promotion of the 3Rs Principles in Teaching & Research (Centro 3R), 56122 Pisa, Italy; 3Department of Pharmacy, University of Genoa, 16132 Genoa, Italy; francesca.musumeci@unige.it (F.M.); grossi@difar.unige.it (G.G.); carbone@difar.unige.it (A.C.); schenone@difar.unige.it (S.S.)

**Keywords:** glioblastoma, Src, pyrazolo [3,4-*d*]pyrimidine scaffold, EGFR

## Abstract

Glioblastoma (GBM) is the most common adult brain tumor and, although many efforts have been made to find valid therapies, the onset of resistance is the main cause of recurrence. Therefore, it is crucial to identify and target the molecular mediators responsible for GBM malignancy. In this context, the use of Src inhibitors such as SI306 (C1) and its prodrug (C2) showed promising results, suggesting that SI306 could be the lead compound useful to derivate new anti-GBM drugs. Therefore, a new prodrug of SI306 (C3) was synthesized and tested on CAS-1 and U87 human GBM cells by comparing its effect to that of C1 and C2. All compounds were more effective on CAS-1 than U87 cells, while C2 was the most active on both cell lines. Moreover, the anti-survival effect was associated with a reduction in the expression of epidermal growth factor receptor (EGFR)^WT^ and EGFR-vIII in U87 and CAS-1 cells, respectively. Collectively, our findings demonstrate that all tested compounds are able to counteract GBM survival, further supporting the role of SI306 as progenitor of promising new drugs to treat malignant GBM.

## 1. Introduction

Glioblastomas (GBM) are malignant and aggressive astrocytic tumors, classified grade IV according to the World Health Organization [1]. They are the most common adult brain tumors, with an annual incidence of about 1/33.330 and with a current median survival of 15 months [2].

Current therapy includes surgery, radiotherapy, and chemotherapy but, unfortunately, in many cases such approaches are ineffective and facilitate the onset of relapse and the development of therapy resistance [3]. Therefore, there is a great need to identify new treatments capable of improving the prognosis of patients. In recent years, several genetic alterations and pathways responsible for GBM malignancy were identified and indicated as targets for new therapies [4,5,6]. In this context, Src-Family Kinases (SFKs), having a key role in the development, tumorigenicity, invasion, and progression of GBM [7], have received particular attention. In this regard, our research group has recently developed new Src inhibitors endowed with a pyrazolo[3,4-*d*]pyrimidine scaffold [8,9,10,11,12,13]. Among this series of compounds, SI306 (compound **1**, C1, Figure 1) showed significant efficacy against U87 human GBM cells [14,15,16]. Based on these successful results, we decided to study this compound further by synthesizing prodrugs potentially endowed with improved pharmacokinetic (PK) properties and testing them on U87 and CAS-1 GBM cells in order to evaluate their efficacy.

In particular, we selected compound **2** (C2, Figure 1), an SI306 prodrug that already showed an improved activity in biological assays [13,16], and the ethylene glycol derivative **3** (C3, Figure 1) as a potential valuable prodrug of a new synthesis. Indeed, a careful analysis of the literature highlighted that introducing a poly(ethylene glycol) (PEG) chain can successfully modulate drug delivery [17]. As different studies pointed out that the activity of many small molecules is improved by PEGylation [18], we synthesized compound **3** (Figure 1), which bears two ethylene glycol units on the carbamate moiety. We chose this short PEG chain driven by the rationale to balance a feasible synthesis with the chance to afford a compound characterized by improved ADME (Absorption, Distribution, Metabolism, Excretion) properties. Compound **3**, and previous analog compounds **1** and **2**, were tested on CAS-1 and U87 cells, and their activities were compared.

The obtained results showed that all tested compounds exhibited a more significant anti-survival effect on CAS-1 than on U87 cells and that C2 was the most active on both cancer cell lines.

Given that Src has been demonstrated to interact and activate epidermal growth factor receptor (EGFR) [19,20], an oncogene frequently amplified and mutated in GBM [21,22,23,24], the action of SI306 (C1) and its derivatives (C2 and C3) on the expression of both wild type (wt) and mutated (vIII) EGFR was evaluated. In this regard, our results showed that CAS-1 cells express EGFR-vIII while U87 cells express EGFR^WT^. Moreover, 48 h exposure to the highest dose (10 μM) of all inhibitors markedly reduced the expression of EGFR^WT^ and EGFR-vIII in U87 and CAS-1 cells, respectively.

Collectively, our findings demonstrate that SI306 and its prodrugs are able to counteract GBM survival by inhibiting Src, as expected. Moreover, for the first time to our knowledge, we have shown that this event is accompanied by a reduction in EGFR^WT^ and EGFR-vIII expression.

## 2. Materials and Methods

### 2.1. Chemistry

All commercially available chemicals were used as purchased from Sigma-Aldrich (St. Louis, MO, USA). DCM was dried over molecular sieves (3 Å, 10% *m*/*v*). TLC was carried out using Merck TLC silica gel 60 F254. Chromatographic purifications were performed on columns packed with Merck silica gel 60, 23−400 mesh, for flash technique. ^1^H-NMR and ^13^C-NMR spectra were recorded on a JEOL JNM ECZ-400S/L1 FT (400 MHz). Chemical shifts were reported relative to tetramethylsilane at 0.00 ppm. Elemental analysis for C, H, N, and S was determined using Thermo Scientific Flash 2000 and results were within ±0.4% of the theoretical value. All target compounds possessed a purity of ≥95% verified by elemental analysis. Compounds **1** and **2** were previously synthesized by us [13,25,26].

### 2.2. Synthesis of 2-(2-hydroxyethoxy)ethyl (3-bromophenyl)(1-(2-chloro-2-phenylethyl)-6-((2-morpholinoethyl)thio)-1H-pyrazolo[3,4-d]pyrimidin-4-yl)carbamate (C3)

A solution of triphosgene (0.45 mmol, 133 mg) in anhydrous CH_2_Cl_2_ (8 mL) was added dropwise to a mixture of NaHCO_3_ (2.25 mmol, 189 mg) and N-(3-bromophenyl)-1-(2-chloro-2-phenylethyl)-6-((2-morpholinoethyl)thio)-1H-pyrazolo[3,4-d]pyrimidin-4-amine 1 (0.45 mmol, 258 mg) in anhydrous CH_2_Cl_2_ (8 mL) precooled at 0 °C in an ice bath. After 30 min, the reaction was allowed to warm to room temperature and stirred for 5 h. Then, a solution of ethylene glycol (20.00 mmol, 1.9 mL) in anhydrous CH_2_Cl_2_ (8 mL) was added and the mixture stirred at room temperature for 16 h. Cold water (25 mL) was added, and the two phases were separated. The aqueous phase was extracted twice with CH_2_Cl_2_ (2 × 25 mL). The organic phase was dried over anhydrous Na_2_SO_4_ and evaporated. The crude oil was purified through two flash chromatographies on silica gel, eluting first with THF/EP 1:1 and then with ethyl acetate/acetone (1:1) affording the desired C3 as a pure light amber oil. Yield 54%. ^1^H-NMR (CDCl_3_): δ 2.48–2.57 (m, 6H, 2CH_2_N morph + NCH_2_CH_2_), 298–3.13 (m, 2H, CH_2_S), 3.47 (t, *J* = 8 Hz, 2H, HOCH_2_CH_2_O), 3.64–3.72 (m, 8H, 2CH_2_O morph + HOCH_2_ + COO-CH_2_CH_2_), 4.41 (t, J = 4 Hz, 2H, COO-CH_2_), 4.68–4.72 and 492–4.98 (2m, 2H, CH_2_N pyraz), 5.48–5.52 (m, 1H, CHCl), 7.16–7.18 (m, 1H Ar), 7.25–7.34 (m, 4H Ar), 7.41–7.43 (m, 3H Ar), 7.49–7.51 (m, 1H Ar), and 7.98 (s, 1H, H-3). ^13^C-NMR (CDCl_3_): δ NMR (101 MHz,) δ = 31.87, 53.45, 53.93, 54.32, 57.66, 60.29, 61.70, 66.58, 66.82, 68.74, 72.62, 76.85, 77.16, 77.48, 103.66, 122.36, 127.44, 127.56, 128.88, 129.17, 130.49, 131.33, 131.93, 135.39, 137.88, 140.98, 153.45, 154.27, 156.07, and 168.36. Anal. calcd. for C_30_H_34_N_6_O_5_SClBr: C 51.03, H 4.85, N 11.90, and S 4.54; found: C 50.77, H 5.19, N 11.57, and S 4.09.

### 2.3. Cell Cultures

U87 and CAS-1 human GBM cell lines were obtained from Ospedale Policlinico San Martino (Genova, Italy). Cells were periodically tested for mycoplasma contamination (Mycoplasma Reagent Set, Aurogene s.p.a, Pavia, Italy). Cells were maintained in DMEM low glucose medium (Euroclone SpA, Pavia, Italy) supplemented with 10% fetal bovine serum (FBS; Euroclone), 2 mM of glutamine (Euroclone).

### 2.4. Treatments

U87 and CAS-1 cells were treated with C1, C2, and C3 for 24, 48, and 72 h with increasing concentrations (1–10 µM) of each compound. The stock solutions of the three compounds were prepared in DMSO (Sigma-Aldrich Co., Saint Louis, MO, USA) and pilot experiments demonstrated that final DMSO doses did not affect any cell response analyzed.

### 2.5. Cell Viability Assay

Cell viability was determined by using the CellTiter 96^®^ AQueous One Solution Cell Proliferation Assay (Promega, Madison, WI, USA), as previously described [27]. Briefly, cells (9 × 10^4^ cells/well) were seeded in 96-well plates (Corning Incorporate, Corning, NY, USA) and then treated. Next, the cells were incubated with MTT solution according to manufacturer’s instructions and the absorbance at 570/630 nm was recorded using a microplate reader (EL-808, BIO_TEK Instruments Inc., Winooski, VT, USA). The relative IC50 values for were calculated by non-linear regression analysis using GraphPad Prism 6.0 software (GraphPad Software, La Jolla, CA, USA).

### 2.6. Blot Analyses

Immunoblots were performed according to standard methods [28] using rabbit antibody anti-human anti-Src (#2109), anti-Phospho-Src (Tyr527, #2105) corresponding in humans to tyrosine 530 residue, anti-Phospho-Src (Tyr416, #2101) corresponding in humans to tyrosine 419 residue, anti PARP (#9542), anti-EGFR (#4267), and anti-EGFR-vIII (#2232; Cell Signaling Technology Inc., Danvers, MA, USA, Upstate, Lake Placid, NY, USA). Anti-rabbit secondary antibody coupled with horseradish peroxidase (Cell Signaling Technologies, Danvers, MA, USA) was utilized.

### 2.7. Statistical Analyses

Data are expressed as means ± S.E.M. Statistical significance of differences was determined by one-way analysis of variances (ANOVA) followed by Tukey’s test. *p* < 0.05 was considered statistically significant.

## 3. Results

### 3.1. Chemistry

Synthesis of C1 and C2 was already reported by us [13,25,26]. C3 was synthesized starting from SI306 with a one-pot two-step procedure. First, trisphosgene was added to a pre-cooled solution of SI306, then ethylene glycol was added, and the reaction was stirred at room temperature affording C3 with good yield (Figure 2).

### 3.2. U87 Cells Are Less Sensitive Than CAS1 to the Effect of SI306- and SI306-Derived Drugs and Show a Major Susceptibility to Compound **2**

In order to compare the effects of C2 and C3 with those of SI306, two human GBM cell lines (U87 and CAS-1) were exposed to increasing concentrations of compounds for 24–72 h.

The analysis of cell viability showed that all compounds, at the same doses and time exposure, induced a major anti-survival effect on CAS-1 cells. In fact, as reported in Table 1, the IC_50_ of all three compounds was lower in CAS-1 than in U87 cells. Moreover, C2 was the most effective on both cell lines, while C3 exerted a lower activity (Table 1).

In detail, SI306 induced a concentration-dependent decrease in the viability of CAS-1 cells, reaching a 40% reduction at 10 μM after 24 h-exposure (Figure 3a, left panel). Instead, no effects on cell viability of U87 were induced by 24 h-treatment with SI306 (Figure 3a, right panel). Analyzing the viability at 48 h, the highest doses of SI306 (8 μM and 10 μM) reduced the viability of CAS1 and U87 by 85% and 25%, respectively (Figure 3b,c, respectively). Moreover, after 72 h, the highest doses decreased the viability of U87 cells by over 70% (Figure 3c), while no further changes were observed in CAS-1 in respect to those observed after 48 h.

With regard to SI306-derived drugs, in both cell lines it has been observed that C2 had a major anti-survival action than SI306 (Figure 3), whilst C3 was less active than SI306 and C2 (Figure 3). In detail, after 24 h-treatment, the highest doses (8 μM and 10 μM) of C2 were able to reduce cell viability more effectively than SI306 (Figure 3b, right panel).

Instead, after 48 and 72 h, the highest doses of C2 and SI306 analogously decreased CAS1 cell viability by 85% and 92%, respectively (Figure 3b,c, respectively, left panels). The viability of U87 cells was significantly impaired after 48 h and, in particular, C2 was more active than SI306 and C3. Interestingly, after 72 h, the highest doses (8 μM and 10 μM) of C2 reduced U87 cell viability by about 90% (Figure 3, right panels).

Moreover, C3 at these highest doses (8 μM and 10 μM) exerted an anti-survival effect on U87 cells comparable to that of SI306, while it was found less effective on CAS1 (Figure 3).

In order to shed light on the kind of cell death, the levels of full length (116 kDa) and cleaved (89 kDa) PARP, a known marker of apoptosis [29], were evaluated. As reported in Figure 4, C1, at 10 μM, and C2 and C3, at 4 and 10 μM, induced apoptosis of U87 cells after 24 and 48 h. The same treatments did not induce PARP cleavage in CAS-1 cells (Figure 4).

### 3.3. SI306 Derivatives Inhibit Src Activity in U87 Cells

Considering that SI306 [25,30] is a Src inhibitor [8,10,11,14,15,16], the inhibitory ability of its derivatives was tested by evaluating the expression levels of Src phosphorylated at Tyr530, a marker of Src inactivation [31], and at Tyr416 (Tyr419 in humans), a marker of Src activation [32]. As shown in Figure 5, in U87 cells, C2 (10 μM) increased the p-Tyr530/Src ratio by 15% and by 20% at 24 and 48 h, respectively (Figure 5, right panels), while C3 (10 μM) induced a 15% increase after 24 h. The analysis of the expression of Src phosphorylated at Tyr416 (Tyr419) showed that 24 h-treatment of U87 cells with all compounds at 4 μM reduced the p-Src/Src ratio by 45% (Figure 5a, left panel). After 48 h, only C1 and C3 reduced the ratio by 20% (Figure 5b, left panel). No significant changes were observed in CAS-1 cells exposed to 4 μM C1, C2, or C3 (Figure 5a,b, left panels). However, 10 μM C1 or C3 reduced the p-Src/Src ratio by 55% in CAS-1 cells treated for 24 h, while only C3 reduced the ratio in U87 cells by 35% (Figure 5a, right panel). After 48 h, C1 and C2 also decreased the p-Src/Src ratio by 30% in U87 cells, while no significant changes were found in CAS-1 cells (Figure 5b, right panel).

### 3.4. CAS-1 Cells Express EGFR-vIII While U87 Cells Express EGFR^WT^ and the Exposure to SI306 or Its Derivatives Has a Different Impact on Their Expression Levels

As it has been demonstrated that progression of GBM is related to Src up-regulation [33] and it is also a consequence of EGFR mutations [34], the expression of both wild type EGFR (EGFR^WT^) and EGFR-vIII, the most common mutant found in GBM, was investigated. Firstly, as shown in Figure 6, EGFR^WT^ was found in U87 while EGFR-vIII was detected in CAS-1 cells.

Notably, 24 h-treatment with 4 μM of C2 or C3 increased the expression of EGFR^WT^ in U87 cells by 10%, while it enhanced EGFR^-^vIII levels in CAS-1 by about 15%, in respect to control cells (Figure 6a, left panels). However, 48 hour-exposure to C3 induced a 30% increase in EGFR-vIII protein levels in CAS-1 cells (Figure 6b, left panel), while 24 h-treatment with 4 μM of SI306 did not significantly change the expression of either EGFR^WT^ or EGFR-vIII in ether cell population (Figure 6a,b, respectively, left panels).

Surprisingly, all compounds at the dose of 10 μM reduced the expression of EGFR-vIII in CAS-1 cells by 70% and 95% after 24 and 48 h, respectively (Figure 6b, right panels), and a 30–35% decrease in EGFR^WT^ protein levels was observed in U87 cells only after 48 h of treatment (Figure 6b, right panels).

## 4. Discussion

Despite ongoing efforts to find valid therapies, GBM remains a highly aggressive tumor with low recovery expectations. In fact, today, the standard cure for GBM is surgery, followed by radiotherapy and/or chemotherapy [2]. However, such approaches are not currently curative, and the onset of resistance is the main cause of recurrence. Therefore, it is crucial to identify the molecular mediators responsible for therapeutic resistance in order to develop targeted strategies capable of overcoming this obstacle. In this regard, several studies have attributed a crucial role to the Src family, both in GBM carcinogenesis and in the mechanisms involved in therapeutic failure [14,35,36]. This evidence suggests that targeting Src could be a valuable approach to improve the life expectancy of GBM patients. Most of the Src inhibitors available today are non-specific and offer cross-reactivity [37]. Moreover, the particular location of GBM makes necessary to develop drugs able to cross the Blood–brain barrier (BBB), and to over-express ATP-binding cassette (ABC) transporters, preventing the drug supply and accumulation in the brain. Many FDA-approved drugs are directed towards these transporters [7] but, although they induce an adequate intake of the drug to the brain, are not effective for GBM treatment [38,39,40].

Therefore, based on these considerations, an effective therapy must hit GBM acting on several fronts and target the molecular pathways involved in the resistance. In recent years, our research team developed an extensive library of pirazolo[3,4-*d*]pyrimidines able to cross the BBB [9,10]. Among these compounds, SI306 (namely C1 in the paper), by inhibiting Src [41,42] and the ABC transporter Pgp [9,10], has demonstrated a powerful antitumor action in vitro and in vivo, both as monotherapy and combined therapy [8,10,13,14]. Similar results were obtained with the prodrug of SI306 (namely C2) [10,13]. Starting from these promising results, a new SI306 derivative (namely C3) was designed and synthesized in order to extend our knowledge on the effect of a different polar chain on the carbamate moiety. C3 was decorated with an ethylene glycol unit in place of the *N*-methyl piperazine fragment present in C2. Both prodrugs were tested on two human GBM cell lines and their effects compared with those of the parent drug SI306. The obtained results herein demonstrate that the anti-survival action of SI306 and its derivatives is both time- and concentration-dependent. In fact, the lowest doses (2 and 4 μM) of all compounds were able to reduce cell survival after 48 h while the highest ones (8 and 10 μM) decreased cell survival already at 24 h with a progressive reduction at 48 and 72 h.

However, although an MTT test revealed that U87 cells are less sensitive than CAS-1, PARP cleavage, as a recognized marker of apoptosis [43], was detected only in U87 treated cells, suggesting that in CAS-1 cells other death pathways occur and further investigation is needed. Interestingly, the pro-apoptotic effect of these compounds on U87 cells is accompanied by Src inactivation measured by evaluating the phosphorylation of Tyr530 and 416 (Tyr419 in humans) [31,32].

With regard to CAS-1 cells, the reduction in cell viability is not related to Src activity and might be due to other mechanisms that deserve further investigation. However, the data demonstrates that Src phosphorylation status does not predict the drug sensitivity in agreement with a recent study reporting that patients with different GBM subtypes expressed similar levels of the unphosphorylated and phosphorylated Src [44]. In the present study, the different sensitivity of two GBM cells to the tested compounds could be related to their pharmacokinetic profile, but also to the fact that CAS-1 cells express EGFR-vIII while U87 cells express EGFR^WT^. This result is in contradiction with a previous data reporting that U87 cells express EGFR-vIII [45] and we believe that this discrepancy could be due to the different specificity of the antibody used in Western blotting analysis.

However, to investigate if all tested compounds are effective on GBM with a different status of EGFR could have important therapeutic implications. In fact, considering that patients with EGFR-vIII have a reduced survival rate [46] and that EGFR-vIII is not expressed in healthy tissues, EGFR-vIII could be considered an excellent target for therapy [47]. To date, drugs developed to treat patients with EGFR-vIII or EGFR have proven to be ineffective due to the inability of such drugs to overcome the BBB, to tumor heterogeneity, and to the activation of compensatory pathways such as Src [21,22,23,24]. Although further investigations are needed, our studies demonstrate that SI306 and its derivatives impact GBM cell survival and that this event is accompanied by a reduction in the expression of EGFR^WT^ and EGFR-vIII in U87 and Cas-1 cells, respectively. These results lead us to believe that these compounds could represent a promising alternative approach to treat GBM.

## Figures and Tables

**Figure 1 pharmaceutics-14-01399-f001:**
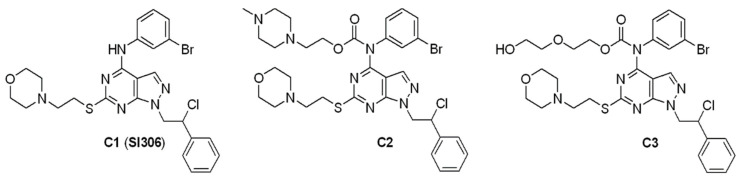
Structures of C1, C2, and C3.

**Figure 2 pharmaceutics-14-01399-f002:**
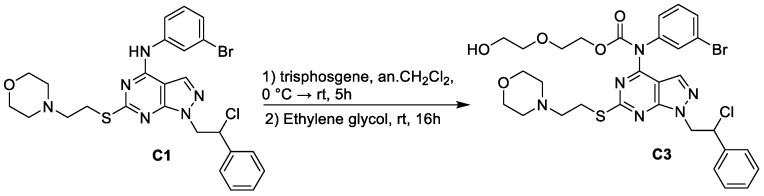
Synthesis of C3.

**Figure 3 pharmaceutics-14-01399-f003:**
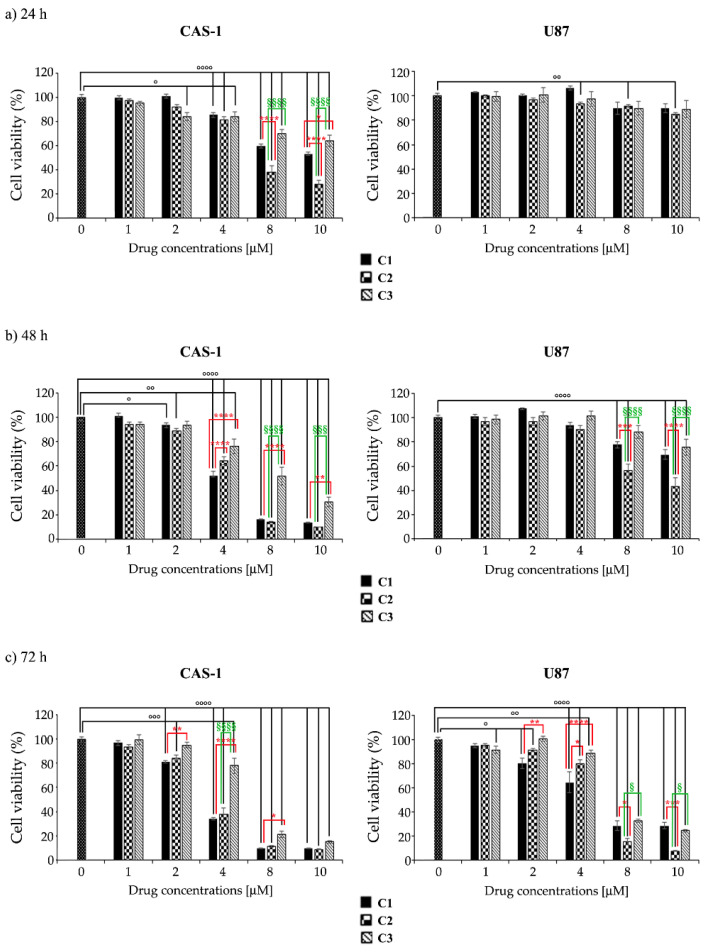
U87 cells are less sensitive than CAS1 to the anti-survival effect of SI306- and SI306-derived drugs. CAS-1 (left panels) and U87 (right panels) cells were exposed to increasing concentrations (1–10 µM) of compound **1** (C1, SI306), compound **2** (C2), and compound **3** (C3) for 24 h (**a**), 48 h (**b**), and 72 h (**c**). Cell viability was evaluated by MTS assay. Histograms summarize quantitative data of the means ± S.E.M. of five independent experiments. Statistical significance of differences was determined by ANOVA followed by Tukey’s test. ° *p* < 0.05 vs. untreated cells; °° *p* < 0.01 vs. untreated cells; °°° *p* < 0.001 vs. untreated cells; °°°° *p* < 0.0001 vs. untreated cells; * *p* < 0.05 vs. compound **1**; ** *p* < 0.01 vs. compound **1**; *** *p* < 0.001 vs. compound **1**; **** *p* < 0.0001 vs. compound **1**; ^§^
*p* < 0.05 vs. compound **2**; ^§§§^
*p* < 0.001 vs. compound **2**; and ^§§§§^
*p* < 0.0001 vs. compound **2**.

**Figure 4 pharmaceutics-14-01399-f004:**
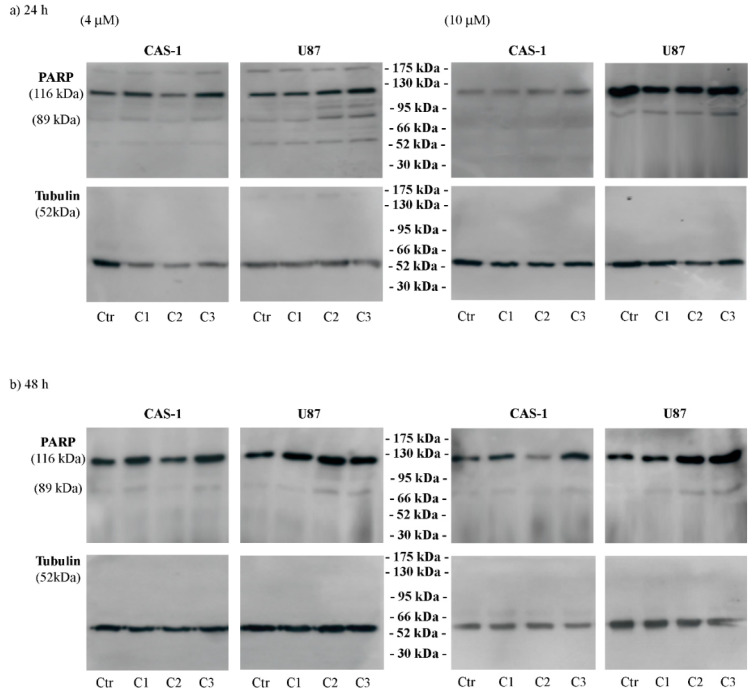
C2 and C3 induce apoptotic death of U87 cells. The expression levels of full length (116 kDa) and cleaved (89 kDa) PARP were evaluated in CAS-1 (left panels) and U87 (right panels) cells exposed to 4 µM and 10 µM of compound **1** (SI306, C1), compound **2** (C2), and compound **3** (C3) for 24 h (**a**) and 48 h (**b**). Immunoblots shown are representative of three independent experiments. Tubulin is the internal loading control.

**Figure 5 pharmaceutics-14-01399-f005:**
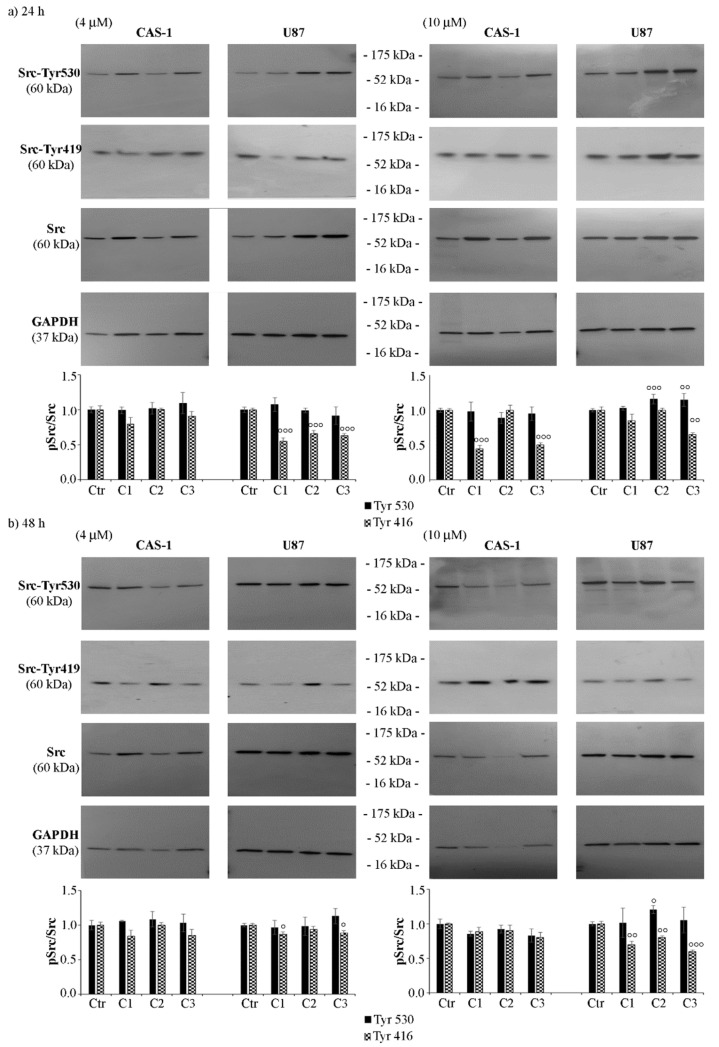
SI306 derivatives inhibit Src activity in U87 cells. Src inactivation was evaluated by immunoblot analyses of p-Src (Tyr530 and Tyr419) and total Src. The expression levels of p-Src and Src were evaluated in CAS-1 (left panels) and U87 (right panels) cells exposed to 4 µM and 10 µM of compound **1** (SI306, C1), compound **2** (C2), and compound **3** (C3) for 24 h (**a**) and 48 h (**b**). Immunoblots shown are representative of three independent experiments. GAPDH is the internal loading control. Histograms summarize quantitative data of means ± S.E.M. of three independent experiments. Statistical significance of differences was determined by ANOVA followed by Tukey’s test. Data are expressed as a ratio of the levels of p-Src to the levels of total Src. ° *p* < 0.05 vs. untreated cells; °° *p* < 0.01 vs. untreated cells; and °°° *p* < 0.001 vs. untreated cells.

**Figure 6 pharmaceutics-14-01399-f006:**
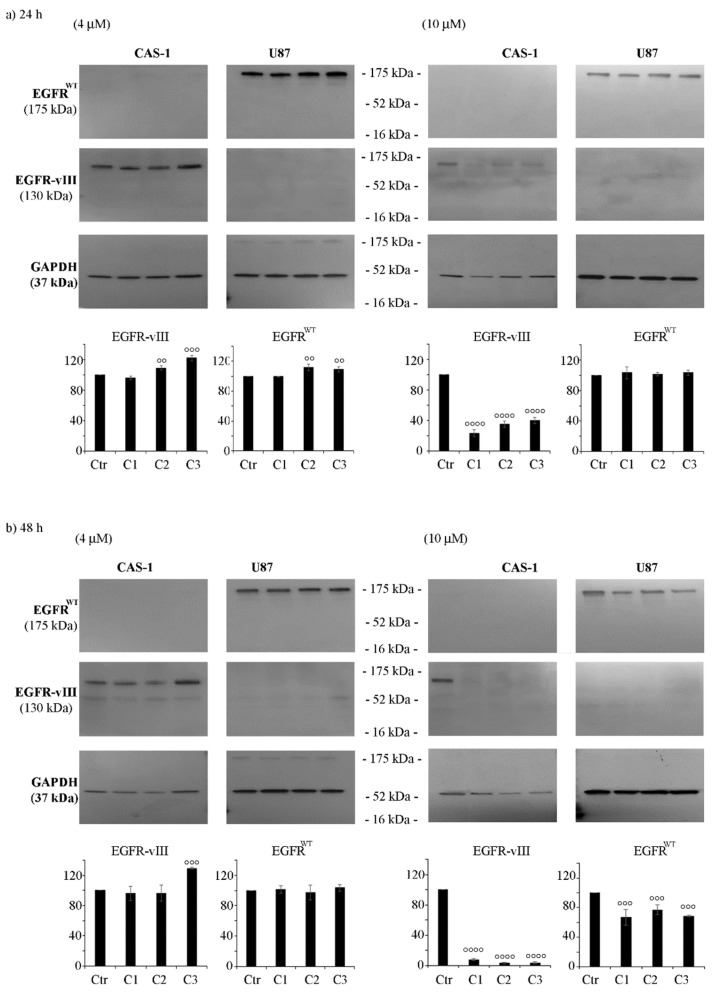
CAS-1 and U87 cells show a different type of EGFR, and treatment with SI306 or its derivatives differently alters their expression levels. The expression of EGFR^WT^ or EGFR-vIII was evaluated by immunoblot analyses performed on protein extracts obtained by CAS-1 (left panels) and U87 (right panels) cells exposed to 4 µM and 10 µM of compound **1** (SI306, C1), compound **2** (C2), and compound **3** (C3) for 24 h (**a**) and 48 h (**b**). Immunoblots shown are representative of three independent experiments. GAPDH is the internal loading control. Histograms summarize quantitative data of means ± S.E.M. of three independent experiments. Statistical significance of differences was determined by ANOVA followed by Tukey’s test. °° *p* < 0.01 vs. untreated cells; °°° *p* < 0.001 vs. untreated cells; °°°° *p* <0.0001 vs. untreated cells.

**Table 1 pharmaceutics-14-01399-t001:** IC_50_ evaluated in CAS-1 and U87 cell lines treated with increasing doses (1–10 μM) of C1 (SI306), C2, and C3 for 24, 48, and 72 h. The data are the means ± S.E.M. of five independent experiments. Statistical significance of differences was determined by ANOVA followed by Tukey’s test. ** *p* < 0.01 vs. C1; *** *p* < 0.001 vs. C1; **** *p* < 0.0001 vs. C1; ^°°°^  *p* < 0.001 vs. C2; and ^°°°°^  *p* < 0.0001 vs. C2.

	CAS-1			U87		
Compound	24 h	48 h	72 h	24 h	48 h	72 h
C1	11.74 ± 0.34	3.88 ± 0.15	3.03 ± 0.1	52.46 ± 1.24	18.63 ± 0.32	5.28 ± 0.35
C2	6.05 ± 0.15 ***	3.66 ± 0.28 ***	3.13 ± 0.16 ***	46.46 ± 0.8 ***	9.24 ± 0.14 ****	3.77 ± 0.15 **
C3	13.92 ± 0.42 ^°°°°^	6.97 ± 0.27 ***^/°°°^	4.47 ± 0.2 ***^/°°°^	54.51 ± 1.16 ^°°°°^	28.36 ± 0.2 ****^/°°°°^	5.79 ± 0.11 ^°°°^

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
