# Peer review of "Anti-Survival Effect of SI306 and Its Derivatives on Human Glioblastoma Cells"

_pharmaceutics, 2022, doi:10.3390/pharmaceutics14071399_

Round 1
Reviewer 1 Report
The study of Lorenzo Monteleone et al aims to evaluate the cytotoxic activity of the Src inhibitor SI306 and 2 derivatives on U87 and Cas-1 glioblastoma cell lines. The results of the study suggest that the compounds more efficiently affect CAS-1 cells viability compared to U87 cells, and that in CAS-1 cells this effect is independent of SRC activity. In contrast, the compounds might slightly induce apoptosis in U87 cells.
The results clearly raise questions on the selectivity of these kinase inhibitors and the molecular basis of their cytotoxic activity, which are a prerequisite to delineate the subgroup of glioblastomas that might respond to these compounds.
I agree that MTT test is widely used for the fast and high throughput analysis of the cytotoxic activity of biological/pharmaceutical compounds. Nevertheless, as I previously mentioned, this assay allows determining the activity of mitochondrial deshydrogenase, and thus the density of viable cells. Changes in the amount of formazan produced –measured by absorbance- might be related to changes in cell metabolism, cell proliferation (cytostatic effect: cell density), or cell death (cytotoxic effect: apoptosis, necrosis).
The Authors mentioned that SI306 and derivatives trigger apoptosis in U87 cells but not in CAS-1 cells. In contrast, the compounds decrease more efficiently the viability of CAS-1 cells as evaluated by MTT assay. This should be discussed.
The impact of the compounds on the apoptosis of U87 cells is very slight. The Authors could have used a positive control, e.g. staurosporine, etoposide. Especially cleaved PARP is observed in untreated U87 cells (Figure 4, upper panel right) and CAS-1 cells (Figure 4 lower panel left). NB the use of at least 2 distinct approaches is often recommended to evaluate apoptosis.
I am not convinced that a 10-20% change in western blot signals even after normalization is reliable. Western blot is a semi-quantitative approach that might be affected by many parameters, including status of the cell culture, cell lysis procedure, sample protein concentration, transfer efficiency and uniformity (among protein size, among samples), sensitivity and linearity of the detection systems (strong vs faint signals), references, normalization, …
It is quite surprising that the histograms of EGFR/ EGFRvIII expression normalized with ponceau S (Figure 4 original version) are perfectly superimposable with those obtained after GAPDH normalization (Figure 6 revised version). The Authors should check they used the right panels.
In human Src, the corresponding tyrosine 416 in avian Src is at position 419. Please, mention this point, and correct the text lines 180, 189, 201 and 288.
Please, provide the reference numbers of the antibodies in the Materials and Methods section.
Author Response
The study of Lorenzo Monteleone et al aims to evaluate the cytotoxic activity of the Src inhibitor SI306 and 2 derivatives on U87 and Cas-1 glioblastoma cell lines. The results of the study suggest that the compounds more efficiently affect CAS-1 cells viability compared to U87 cells, and that in CAS-1 cells this effect is independent of SRC activity. In contrast, the compounds might slightly induce apoptosis in U87 cells.
The results clearly raise questions on the selectivity of these kinase inhibitors and the molecular basis of their cytotoxic activity, which are a prerequisite to delineate the subgroup of glioblastomas that might respond to these compounds.
I agree that MTT test is widely used for the fast and high throughput analysis of the cytotoxic activity of biological/pharmaceutical compounds. Nevertheless, as I previously mentioned, this assay allows determining the activity of mitochondrial deshydrogenase, and thus the density of viable cells. Changes in the amount of formazan produced –measured by absorbance- might be related to changes in cell metabolism, cell proliferation (cytostatic effect: cell density), or cell death (cytotoxic effect: apoptosis, necrosis).
- The Authors mentioned that SI306 and derivatives trigger apoptosis in U87 cells but not in CAS-1 cells. In contrast, the compounds decrease more efficiently the viability of CAS-1 cells as evaluated by MTT assay. This should be discussed.
As required, this issue has been discussed in the text (lines 258-261).
- The impact of the compounds on the apoptosis of U87 cells is very slight. The Authors could have used a positive control, e.g. staurosporine, etoposide. Especially cleaved PARP is observed in untreated U87 cells (Figure 4, upper panel right) and CAS-1 cells (Figure 4 lower panel left). NB the use of at least 2 distinct approaches is often recommended to evaluate apoptosis.
We agree with the reviewer that the use of at least 2 distinct approaches is often recommended to evaluate apoptosis. However, due to the limited time for the revision (5 days) and considering that in this moment other tools to evaluate apoptosis (e.g. Annexin-V labeling or bax/bcl2 antibodies) are not disposable in our lab, we cannot now satisfy this request.
- I am not convinced that a 10-20% change in western blot signals even after normalization is reliable. Western blot is a semi-quantitative approach that might be affected by many parameters, including status of the cell culture, cell lysis procedure, sample protein concentration, transfer efficiency and uniformity (among protein size, among samples), sensitivity and linearity of the detection systems (strong vs faint signals), references, normalization, …
We agree with the Reviewer that the western blot (WB ) is a semi-quantitative approach that might be affected by several parameters. In order to limit the impact of these factors, WB analyses have been carried out for three independent experiments on protein samples obtained from cells, at the same culture passage and at the same density. Notably, the procedure of cell lysis was the same for all samples and protein concentration was evaluated using the same method (Pierce TM BCA Protein Assay Kit, Thermo Scientific TM). Moreover, in order to be sure that the transfer efficiency and the uniformity are adequate, all membranes, after every transfer, were stained with Red Ponceau and only those that showed uniformity of signals (similar amount of protein loading) were used for subsequent WB analyses. The signals of WB were detected by incubating membranes with the ECL substrates (Pierce™ ECL Plus Western Blotting Substrate): this is a method widely used by world’s scientific community. Then, every membrane was incubated with antibodies against GAPDH and/or Tubulin used as loading control.
- It is quite surprising that the histograms of EGFR/ EGFRvIII expression normalized with ponceau S (Figure 4 original version) are perfectly superimposable with those obtained after GAPDH normalization (Figure 6 revised version). The Authors should check they used the right panels.
The panels shown in figures 4 and figure 6 are correct. In fact, due to the reasons explained at the point 3, the histograms reported (protein normalized with GAPDH), which are an average of three independent experiments, were superimposable with data obtained by Red Ponceau normalization.
- In human Src, the corresponding tyrosine 416 in avian Src is at position 419. Please, mention this point, and correct the text lines 180, 189, 201 and 288.
As required, the tyrosine residue number of Src has been properly corrected.
- Please, provide the reference numbers of the antibodies in the Materials and Methods section.
As required, the reference numbers of the antibodies were provided in the Materials and Methods section.
Reviewer 2 Report
The authors of Anti-survival effect of SI306 and its derivatives on human glioblastoma cells described the use of a Src inhibitor and its derivatives to control the growth of Glioblastoma cells.
While the manuscript is well written and designed, some clarification and revisions are required as per below:
1- Please comment on the use of different internal loading controls (tubulin vs GAPDH vs ponceau). The choice of different loading controls is usually related to different expression levels and consistency within different tissues but for the purpose of these experiments the same cell lines are used through out the work.
2-Figure 5b and Figure 6 shows absence or minimal protein expression in the GAPDH in CAS-1 cells with the C2 compound. Is C2 altering the expression or stability of GAPDH in this cell line? It is unsafe to draw conclusions regarding protein expression of target proteins (SRC and EGFR in this case) when the loading control is drastically altered. I would like to see a blot showing another loading control (preferably tubulin) or normalizing against loading control as well (which can be hard in the samples where the protein GAPDH is absent).
Author Response
The authors of Anti-survival effect of SI306 and its derivatives on human glioblastoma cells described the use of a Src inhibitor and its derivatives to control the growth of Glioblastoma cells.
While the manuscript is well written and designed, some clarification and revisions are required as per below:
- Please comment on the use of different internal loading controls (tubulin vs GAPDH vs ponceau). The choice of different loading controls is usually related to different expression levels and consistency within different tissues but for the purpose of these experiments the same cell lines are used through out the work.
We agree with the Reviewer's comment regarding the use of different internal loading controls. As the reviewer has seen, we have properly deleted Ponceau and used tubulin to normalize PARP levels. The reason why we have preferred to use GAPDH, to normalize Src and EGFR, is due to the fact that the signals of src, being about 60 kDa, could interfer with those of tubulin (52 kDa).
- Figure 5b and Figure 6 shows absence or minimal protein expression in the GAPDH in CAS-1 cells with the C2 compound. Is C2 altering the expression or stability of GAPDH in this cell line? It is unsafe to draw conclusions regarding protein expression of target proteins (SRC and EGFR in this case) when the loading control is drastically altered. I would like to see a blot showing another loading control (preferably tubulin) or normalizing against loading control as well (which can be hard in the samples where the protein GAPDH is absent).
The choice of GAPDH as the loading control has been explained in the previous point.
However, in order to satisfy the Reviewer's request, the same membranes have been incubated with anti-tubulin antibody and the results were shown in supplementary materials.
Moreover, the histograms shown in Figure 5 report the ratio of the phosphorylated and non-phosphorylated forms of Src that were normalized on the GAPDH. Notably, since the denominator of each ratio (GAPDH) is the same, the ratio of the phosphorylated and non-phosphorylated forms is not affected by the expression levels of the loading control.
Reviewer 3 Report
The manuscript can be accepted now.
Author Response
Comments and Suggestions for Authors
The manuscript can be accepted now.
We thank the Reviewer for appreciating the work done to satisfy all requests and comments.
This manuscript is a resubmission of an earlier submission. The following is a list of the peer review reports and author responses from that submission.
Round 1
Reviewer 1 Report
Dear Editors,
The Study of Dr Monteleone et al aims to evaluate the cytotoxic activity of the Src inhibitor SI306 (compound 1) and 2 prodrug derivatives (compounds 2 & 3) on the U87 and Cas-1 glioblastoma cell lines. The Authors report that these 3 compounds reduced cell viability, inhibited Src in U87 cells, and decreased EGFR expression and phosphorylation -compound 2 being the more active-. This study is of importance considering the frequent activation of Src and EGFR in glioblastoma, and the aggressiveness and the poor prognosis of this cancer. Nevertheless as a biologist, I consider that the Authors could have gone deeper in the analysis of the cytotoxic activity of these compounds.
Sincerely yours
Authors
The Study of Dr Monteleone et al aims to evaluate the cytotoxic activity of the Src inhibitor SI306 (compound 1) and 2 prodrug derivatives (compounds 2 & 3) on the U87 and Cas-1 glioblastoma cell lines. The Authors report that these 3 compounds reduced cell viability, inhibited Src in U87 cells, and decreased EGFR expression and phosphorylation -compound 2 being the more active-. This study is of importance considering the frequent activation of Src and EGFR in glioblastoma, and the aggressiveness and the poor prognosis of this cancer. Nevertheless, the manuscript raises the following issues.
The title of the manuscript is a little bit misleading. There is no direct proof that SI306 impaired glioblastoma cell survival through reducing EGFR expression.
CAS-1 cells are much more sensitive to SI306 derivatives than U87 cells. In contrast, these compounds do not inhibit Src in CAS-1 cells, but decrease Src activity in U87 cells. This raises concerns about the selectivity of these inhibitors, their molecular targets and the mechanism of their cytotoxic activity.
MTT test allows determining the density of viable cells. The Authors could have completed their study by evaluating cell proliferation and apoptosis.
The decrease in EGFR expression following SI306 treatment was previously reported (Ref 16 of the manuscript). Nevertheless, based on the differences in sample loading as evidenced with Ponceau staining and Src immunoblots in Figures 3 and 4, I am not convinced that a 10-20% change in EGFR or in phospho-Src (Tyr530) signals after normalization is reliable. In this context, the Authors do not discuss the increased EGFR expression in CAS-1 cells treated with 4uM compounds 2/3, whereas a dose of 10uM down-regulated the receptor.
The Authors could have evaluated the mechanism involved in the down-regulation of EGFR. Does the compounds affect EGFR at mRNA levels, protein stability, EGFR internalization/recycling, receptor shedding ? This process seems independent of Src. Which epitopes are recognized by the antibodies ? The Authors should provide the corresponding references. The Authors could have used more quantitative assays, e.g. measurement of Src activity. Especially, the phosphorylation of tyrosine 419 (corresponding to the site 416 in avian Src) is critical for Src activity. They could have evaluated downstream pathways, e.g. AKT assay.
The anti-phospho Src antibody has been raised against the avian Src phosphorylated at position 527. In human Src, the corresponding tyrosine is at position 530. Please, mention this point in line 289, and correct the text lines 131, 140, 213.
The Authors should indicate the post-hoc test used following ANOVA.
They should provide the SEM of the IC50 in the Table. Some IC50 values are highly extrapolated (U87 cells, 24h treatment) according to the higher dose used. How these values have been determined ?
Line 152 Reference 28 does not mention that EGFR mutations trigger Src up-regulation.
Please, check the text for misprints, e.g. line 129 “inibitory”
Reviewer 2 Report
Dear authors,
Thanks for your efforts on doing the hard work and on a well written manuscript. Following are some recommendations for a better version of your findings:
1- Title: SI306 and its prodrugs counteract glioblastoma cell survival by 2 reducing EGFR expression. The title implies a direct action of the drugs on EGFR expression and a reduction of cell survival of glioblastoma cell survival. This would be more acceptable if you established that link through siRNA and knockdown of EGFR or other silencing tools. Maybe overexpression of EGFR by a vector and then trying out the drugs can justify the current title
The way the current findings stand I see EGFR expression as one of the mechanisms of many other unexplored here mechanisms which these drugs are inducing. I would suggest
SI306 and its prodrugs counteract glioblastoma cell survival and is associated with a reduction in EGFR expression
2- Table 1: Can you comment on how the IC50 was calculated? please refer to the following to identify whether your values represent IC50 or EC50 and whether they are absolute or relative (PMID: 22328315 DOI: 10.1002/pst.426)
3- Authors mention cytotoxic effect when commenting on MTT experiments which is meant to measure viability and proliferation (intact cells dividing or not). Cytotoxicity is usually measure as LDH is released outside the cell and a different assay measures that. I suggest carefully using the word cytotoxic which was measured here and be replaced by inhibitory effect or anti-proliferative effect.
4- Measuring SRC kinase activity:
Result 2.3. SI306 derivatives inhibit Src activity in U87 cells
Promising data but strong statement to be inferred from simply measuring expression of Src 527. Src 416 is a better measurement of active Src and a inhibition of active SRC can be addressed with an kinase assay
5-For western blots, Ponceau staining is a good choice but would require a house keeping gene western blot to normalize data in figure 4
6- Figure 1 uses compound 1-2-3 whereas all other figures use C1-C2-C3. I would recommend using C1-2-3 throughout
Reviewer 3 Report
Dear authors,
Thanks for your manuscript entitled" SI306 and its prodrugs counteract glioblastoma cell survival by reducing EGFR expression'' This manuscript is written in general. But before acceptance in this reputed journal it needs some major modifications.
Comments and suggestions
- Please make your abstract more attractive for the readers. In the abstract, line 17 has written SI306 is compound 1. however, line 18 has written compound 3. Please check and confirm.
- Line 16, In this context is it right?
- please indicate your specific aim in the abstract as well as the end of the introduction.
- Line 31-32, please check this sentence has been written correctly.
- Line 41-42 SI306 why underline?
- Line 59, EGFR, if you used the first time please abbreviate it.
- Some places write 24 hours and some places write 24 hour, please make it consistent.
- Why authors choose (8 M and 10 M) and 24, 48, and 72 hours. It's time-dependent or concentration-dependent.
- Which test was performed for analyzing your results that should be mentioned in each figure legend in the results section.
Reviewer 4 Report
In this manuscript, the authors conducted research on “SI306 and its prodrugs counteract glioblastoma cell survival by reducing EGFR expression”. The results are not rich enough to support their conclusions. The manuscript is recommended to be rejected. Specific questions.
- the number of compounds designed and synthesized by the authors is small and the structure is single. Compound activity is not good enough.
- Src, EGFR regulates multiple cell behaviors, and the MTT assay alone is not sufficient to detect the effects of the compounds on cells.
- most of the results in which compounds affect cellular proteins are not significant.
- usually we use tubulin, gapdh, actin as loading control for wb